# Impact of a 10-Week Aqua Fitness Intervention on Physical Fitness and Psychosocial Measures in Inactive Healthy Adult Women

**DOI:** 10.3390/healthcare13030334

**Published:** 2025-02-06

**Authors:** Athanasios A. Dalamitros, Aristotelis Kouloglou, Giorgos Nasoufidis, Kleopatra Stogiannidou, Nur Eradli, Vasiliki Manou

**Affiliations:** 1Laboratory of Evaluation of Human Biological Performance, Department of Physical Education and Sport Science, Aristotle University of Thessaloniki, 57001 Thessaloniki, Greece; akoulog@phed.auth.gr (A.K.); gnaso@phed.auth.gr (G.N.); kleopatrag@gmail.com (K.S.); vmanou@phed.auth.gr (V.M.); 2Department of Management, Graduate School, Istanbul Technical University, Istanbul 34367, Türkiye; eradli23@itu.edu.tr

**Keywords:** aquatic-based exercise program, fitness profile, psychosocial functioning, intervention, short cessation period, women

## Abstract

**Background/objectives**: Previous studies on aquatic exercises have primarily focused on either physical fitness or psychological outcomes. This study examines the effects of a structured 10-week aqua fitness program on physical fitness and psychosocial outcomes in healthy adult women. Additionally, a 4-week training cessation period was incorporated to assess the sustainability of any observed physical fitness benefits. **Methods**: A total of 32 female participants (mean age 51.28 ± 9.12 years) with prior aqua aerobics experience engaged in supervised aqua fitness sessions, conducted three times per week (~55 min/session) at moderate intensity (RPE = 12, on a 6–20 scale). The physical fitness outcomes assessed included dominant hand grip strength, lower limb muscle endurance, dynamic balance, mobility, and upper and lower limb flexibility. The psychosocial outcomes included subjective well-being and social inclusion. **Results**: The results demonstrate significant improvements in dynamic balance (ES = 0.85) and lower limb flexibility (ES = 0.73 and 0.65 for the two limbs, respectively), with smaller yet notable gains observed in other physical fitness outcomes (ES = from 0.20 to 0.48). On the contrary, only a marginal improvement was detected in a single domain of subjective well-being (*environmental* domain, ES = 0.35) and no changes were observed across the seven domains of social inclusion. Importantly, all physical fitness gains were maintained during the 4-week training cessation period, with lower limb flexibility showing additional improvements. **Conclusions**: These findings underscore the effectiveness of supervised aqua fitness programs in enhancing physical fitness in middle-aged women, while their impact on psychosocial outcomes appears limited in this population.

## 1. Introduction

Physical activity is widely recognized as a cornerstone of overall health and well-being [1], with particular importance for women, who often encounter unique physiological and lifestyle-related challenges [2,3]. Among the diverse exercise modalities available, aquatic-based fitness programs, commonly referred to as aqua fitness, have gained considerable traction in recent years due to their low-impact but highly effective nature, which aligns with fundamental health and fitness objectives [4,5]. Evidence suggests that regular participation in aqua fitness programs can significantly improve cardiovascular fitness, muscular strength, flexibility, and body composition, comparable to land-based exercise programs [6]. These programs incorporate exercises performed in water, utilizing its distinctive properties, such as buoyancy and natural resistance, which provide a joint-friendly yet rigorous workout, especially during lower limb exercises [7]. The aquatic environment minimizes the risk of injury, enhances circulation, and supports a greater range of motion, making aqua fitness accessible to individuals across a wide spectrum of ages and fitness levels [8]. Furthermore, aqua fitness represents an optimal exercise option for women managing age-related health conditions, including osteoporosis [9], osteoarthritis [10], hypertension [11], and other musculoskeletal issues [12].

The physical benefits of structured aqua fitness programs for women across different age groups are well-established; however, their potential impact on social functioning (i.e., a person’s interaction with their environment and their ability to fulfill roles such as work, social activities, and relationships with family) [13] and quality of life warrants equal attention. Previous review studies [14,15] have highlighted that participation in aqua fitness programs can enhance quality of life and mood while alleviating symptoms of depression, resulting in significant mental health improvements. These benefits are often accompanied by increased social interaction and reduced stress, which are critical for sustaining long-term health and well-being [16]. Also, group-based aquatic activities have been shown to foster positive life satisfaction, further contributing to participants’ overall well-being [17]. Although the priorities for engaging in such programs may differ across age groups—for example, older women tend to prioritize functional fitness and health [18], while younger women often focus on body composition and esthetics [19]—mood and social connectedness improvements have been observed consistently in both younger and older participants [20]. These findings underscore the multifaceted benefits of aqua fitness programs, extending beyond physical health to encompass psychosocial and emotional dimensions. Still, these aspects are more challenging to quantify compared to physical fitness measures [21].

To the author’s knowledge, previous studies have primarily focused on the effects of aqua fitness programs on either physical fitness [8] or psychosocial measures in healthy female individuals [22], without comprehensively addressing both dimensions. As an exception, it is only the intervention study of Fail et al. [23], in a sample of older women, that included the quality of life measure in its experimental design. With this in mind, the current study aims to investigate specific physiological improvements and changes in psychosocial functioning associated with participation in aqua fitness, confirming its potential as an effective and enjoyable strategy for women seeking to enhance overall health and well-being. Additionally, the inclusion of a detraining period allowed for an examination of the sustainability of these positive changes, offering insights into the long-term impact of regular physical activity and its role in promoting adherence. Hence, our primary objective was to assess the effects of a structured and supervised 10-week aqua fitness program, followed by a 4-week detraining period, on health-related fitness parameters and psychosocial functioning in healthy women across different age groups. Based on the findings of previous studies, the following hypotheses were formulated: (i) participation in a 10-week structured aqua fitness program would result in significant improvements in functional fitness and psychosocial outcomes, and (ii) the physical fitness adaptations achieved during the intervention would diminish following the 4-week cessation period.

## 2. Materials and Methods

### 2.1. Participants

This quasi-experiment study utilized a sample size calculation conducted before the exercise intervention through power analysis, using the statistical G*Power software package for repeated measures and analysis of variance (ANOVA) (GPower, v.3.1.9, University of Kiel, Kiel, Germany). This analysis determined that a sample size of 30 participants would provide a statistical power of 0.80 and an effect size of 0.25 to detect meaningful effects, with a significance level set at *p* < 0.05. Ultimately, 32 female participants (mean chronological age 51.28 ± 9.12 years; height 165.80 ± 0.05 cm) were recruited. The convenience sampling method was used, selecting participants who had previously engaged in aqua training classes within the same residential area, prior to the six-month seasonal closure of the municipal swimming pool during winter. All participants obtained medical clearance from a primary healthcare professional before enrollment. The inclusion criteria were as follows: (i) aged between 30 and 60 years; (ii) physically healthy status at baseline; (iii) absence of regular physical activity during the six months preceding the intervention; (iv) a minimum of six months experience in water fitness programs; and (v) no history of musculoskeletal or neurological injuries, conditions, or syndromes diagnosed within the previous four months. Participants who failed to meet a minimum compliance rate of 80% attendance in the training program were excluded from the analysis. During the 4-week training cessation period, participants were instructed to refrain from structured physical exercise, but were encouraged to continue their usual physical activities (e.g., walking, sea swimming) and maintain habitual dietary practices. All participants were fully informed about the study’s potential benefits and risks before voluntarily providing written informed consent. The study procedures adhered to the principles outlined in the Declaration of Helsinki and were approved by the Institutional Review Board (approval number: 5012/4 March 2024).

The parameters evaluated in this study were categorized into two main groups: physical fitness and psychosocial outcomes. These parameters were assessed at three time points: at baseline, before the commencement of the 10-week program (training week 0: T0), after the program (training week 10: T10), and during the fourth week of the cessation period (detraining week 4: D4). It is important to note that during the D4 measurement, only physical fitness parameters were assessed. To familiarize participants with the testing procedures, a session was conducted from 24 to 48 h prior to the start of the training program. All measurements were carried out before the aqua fitness session, lasting approximately 15 min. Assessments were performed individually in a room within the swimming pool facilities, specifically adapted for this purpose. Throughout the course of the exercise intervention, participants were instructed to maintain their usual physical activity and dietary routines. The following sections provide a detailed description of the physical fitness and the psychosocial measures employed in this study.

### 2.2. Physical Fitness Parameters

Chair sit–and–reach test: The sit–and–reach test is a validated measure for assessing lower limb flexibility, with an intraclass correlation coefficient range of 0.98 [24]. Participants were seated on an adapted chair that did not meet the standard height requirement, with both knees fully extended and heels resting on the floor. Participants were instructed to inhale deeply and, while exhaling, reach forward as far as possible toward the extended foot, first on the right side and then on the left, ensuring that both hands and middle fingers were aligned. Each side was tested twice, and the maximum value achieved for each side was recorded.

The back scratch test is a reliable measure of upper limb flexibility, with an intraclass correlation coefficient of 0.96 [24]. In this test, participants were instructed to reach behind their head and down their spine with their right hand, while simultaneously reaching behind their back and up the spine with their left hand, ensuring that both middle fingers were aligned toward the center of the back. Participants were allowed one additional attempt with the same hand, and the score was recorded based on the distance (in centimeters) between the middle fingers, with the maximum value noted. The procedure was then repeated for the opposite hand.

Timed up-and-go test: Mobility was measured using the timed up-and-go (TUG) test, which has demonstrated high reliability, with an intraclass correlation coefficient range of 0.93–0.98 [25]. Participants began seated on a standard chair without armrests, with their upper limbs crossed at shoulder level and placed against the chest. Upon the signal, participants were instructed to stand up and walk as quickly as possible to a cone positioned 8 feet (244 cm/96 inches) in front of the chair, without running. They were required to perform a swift turnaround without pausing and then return to a seated position in the chair. One trial was conducted, and the time taken to complete the task was recorded using a handheld stopwatch.

Handgrip dynamometer: The upper limb strength of the dominant arm was assessed using a handgrip dynamometer, which has demonstrated high reliability, with an intraclass correlation coefficient of 0.98 [26]. Participants stood up, holding the handgrip dynamometer with their right hands extended downwards. After relaxing their shoulders, they were instructed to squeeze the handgrip for three seconds. Each hand was tested twice, and the maximum value achieved was recorded. Performance was measured in kilograms (kg).

The 30 s sit-to-stand test: The 30 s sit-to-stand test validates lower limb muscle endurance, with an intraclass correlation coefficient of 0.71 [27]. Participants were seated on a standard chair without armrests, with their upper limbs crossed at shoulder level and placed against their chest. Upon receiving the signal, participants were instructed to stand up from the chair and sit back down as quickly as possible, performing the movements continuously. Each time they stood up; their knees were required to be fully extended. The maximum number of repetitions was recorded, with participants completing the test only once.

Reach forward test: The reach forward test evaluated dynamic balance. A tape measure was positioned at the participants’ shoulder height, and they were instructed to extend their right hand forward, reaching as far as possible until they felt that they had lost their balance. Each participant was allowed only one attempt, and the distance reached, measured in centimeters, was recorded. The intraclass correlation coefficient for this test had a range of 0.90 to 0.97 [28].

### 2.3. Psychosocial Parameters

Two self-report take-home questionnaires, adapted for Greek, were used to evaluate participants’ psychosocial parameters. More analytically, the *WHO Quality of Life Instrument-Short Form* (WHOQOL-BREF; Skevington et al., 2004 [29]) was used, which includes 24 items covering various domains of subjective quality of life, such as physical health, psychological health, social relations, and environment, along with 2 items related to the general quality of life and general health. This instrument is appropriate for use with healthy populations, with an internal consistency reliability value of 0.92 (Cronbach’s alpha value) [30]. The second questionnaire, focused on social inclusion, consisted of 19 questions categorized into 8 domains (i.e., social capital bonding, social capital bridging, social acceptance, neighborhood cohesion, engagement in leisure and cultural activities, citizenship, stability of housing tenure, and security) [31]. Previous research has indicated good internal consistency for this measure (Cronbach’s alpha value: 0.80) [32]. In both questionnaires, participants rated their responses using a 5-point Likert scale (ranging from 1, indicating “very poor”, to 5, indicating “very good”). Participants were asked to return both completed questionnaires during their next visit to the swimming pool.

### 2.4. Training Program

The training program lasted for 10 weeks, with sessions held three days per week (from May to July), spaced 24 h apart, and conducted at the same time of day (from 9.00 to 10.00 A.M.) in a 50 m outdoor pool with a uniform depth of 1.80 m in the area of “Alana Toumba” in Thessaloniki, Greece. The intensity of the training was monitored using the Rating of Perceived Exertion (RPE) scale (Borg scale, ranging from 6 to 20) on the final day of each week’s training session, with an intensity level maintained at approximately RPE = ~12 throughout the entire 10-week period. This index provides a simple and accessible method for participants to assess their perceived exertion. It is considered particularly useful in aqua fitness settings, where heart rate monitoring may be less feasible due to the aquatic environment [33]. In the 6th week of the program, the duration of each session was increased by 10 min. The training cessation period coincided with the vacation period in August. The mean water temperature during the program ranged from 26.8 to 27.2 °C, and the air temperature varied between 28.0 and 32.0 °C. The pool depth remained constant at 1.80 m. The program consisted of a combination of exercises designed to target endurance, muscle strength, and flexibility, using a circuit-based format (Table 1). A 3 min warm-up, focusing on muscular activation, preceded each training session. Each session was limited to a maximum of 17 participants. The same two researchers specialized in aqua training, strength, and conditioning programs supervised all training sessions and testing procedures. The experimental design is demonstrated in Figure 1.

#### Statistical Analysis

Statistical analyses were performed using the SPSS statistical package (v. 28.0). To analyze the impact of the entire evaluation period (from T0 to D4) on physical fitness parameters, including upper and lower limb flexibility, muscle strength, mobility, and dynamic balance, a two-way repeated measures analysis of variances (ANOVA) was performed. Paired samples *t*-tests with the Bonferroni correction were utilized to examine within-group changes in psycho-social measures (from T0 to T10) and fitness parameters (from T0 to T10, T0 to D4, and T10 to D4). The effect size (ES) was determined using eta-squared (*η*^2^), with the following classifications: trivial (<0.19, small (0.20–0.59), moderate (0.60–1.19), large (1.20–1.99), and very large (2.0–4.0). Additionally, 90% confidence intervals were calculated for changes between T0 and T10 as well as T10 and D4. A significance level of a = 0.05 was adopted.

## 3. Results

According to two-way repeated measures (ANOVA), no significant effect of the period evaluated (from T0 to D4) was noticed for body weight (*F*_2,56_ = 2.209, *p* = 0.136, *η*^2^ = 0.73). Conversely, upper body muscle strength of the dominant arm (*F*_2,56_ = 4.562, *p* = 0.034, *η*^2^ = 0.140), lower limb muscle endurance values (*F*_2,56_ = 5.211, *p* = 0.008, *η*^2^ = 0.157), dynamic balance (*F*_2,56_ = 18.724, *p* < 0.001, *η*^2^ = 0.401), and mobility scores were improved at D4 (*F*_2,56_ = 17.180, *p* < 0.001, *η*^2^ = 0.380). The same was noticed for the flexibility scores of both the upper (*F*_2,56_ = 10.503, *p* < 0.001, *η*^2^ = 0.273, *F*_2,56_ = 3.596, *p* < 0.05, *η*^2^ = 0.114, for the right and the left arm, respectively) and lower limbs (*F*_2,56_ = 16.327, *p* < 0.001, *η*^2^ = 0.368, *F*_2,56_ = 21.925, *p* < 0.001, *η*^2^ = 0.439, for the right and the left arm, respectively). Paired sample *t*-tests showed improvements in upper-body muscle strength between T0 and T10 (*p* < 0.001), lower limb muscle endurance scores between T0 and T10 (*p* = 0.019), and dynamic balance scores between T0 and both T10 and D4 (*p* < 0.001). Similarly, mobility scores improved from T0 to T10 and D4 (*p* = 0.006, *p =* 0.031). The flexibility scores of the right arm improved between T0 and both T10 (*p* = 0.003) and D4 (*p* = 0.004), while for the left arm, improvements were observed from T0 to D4 (*p* = 0.031). Finally, the flexibility scores of both legs were improved between all testing points (*p* < 0.001–0.028) (Table 2). The analysis of the WHOQOL-BREF questionnaire revealed significant improvements only in the *environmental* domain (*t* = −2.61, *p* = 0.040, ES = 0.35). Finally, none of the seven domains of the social inclusion questionnaire were altered significantly.

## 4. Discussion

This study investigates the effect of a 10-week structured and supervised aqua fitness program, followed by 4 weeks of training cessation, on upper-body muscle strength, lower-body muscle endurance, balance, mobility, and flexibility of the upper and lower extremities. Additionally, this study examines the impact of this intervention on quality of life and social inclusion. The study sample consisted of physically healthy women aged 30–60 with prior experience in similar fitness programs. The results indicate significant improvements in physical fitness parameters following the 10-week aqua fitness program. Notably, these gains were preserved during the subsequent 4-week period of training cessation. However, no significant changes were observed in quality of life and social inclusion domains after the training period, except for the *environmental* domain, which showed a positive effect on subjective well-being. Based on these findings, the first hypothesis—that 10 week of aqua fitness sessions (conducted three times per week for 50 to 60 min each) would provide a sufficient stimulus to enhance performance in functional fitness tests and maintain these adaptations during a 4-week training hiatus—was supported. In contrast, the second hypothesis—that the intervention would result in notable improvements in psychosocial functioning—was largely rejected, as only minor enhancements were observed.

The lack of significant improvements across a broad spectrum of social functioning, as measured by the social inclusion questionnaire in this study, can be attributed to the complex and multifaceted nature of social relationships [34]. Social functioning is influenced by individual characteristics and external life circumstances, such as career stability and life transitions (e.g., housing and marital status) [35]. These factors are generally resistant to change, particularly during middle adulthood, as individuals establish stable social behaviors and report greater emotional satisfaction with their social lives [36]. According to Wrzus et al. (2013) [37], social interactions tend to plateau around the age of 30 and remain stable into later adulthood (45+), which aligns with the age range of the participants in the study. Additionally, the reliance on self-reported questionnaires to analyze within-person variations may lack the necessary sensitivity to detect subtle changes in social functioning, as previously noted [38]. Moreover, subjective well-being, as evaluated using the WHOQOL-BREF questionnaire, appears to interact with social functioning in a complex way. While prior studies have demonstrated a significant positive relationship between social activity and subjective well-being [39,40], the quality of social interactions (e.g., time spent with family members) is more critical than the quantity (e.g., participating in group events) when evaluating subjective well-being [41]. Furthermore, the diversity in definitions and measurement tools for constructs such as social functioning and subjective well-being introduces challenges in generalizing and interpreting results across studies [42]. For instance, in the study of Ayan et al. (2017) [43], a 6-month water-based exercise intervention improved the quality of life among healthy adult women (aged 46.5 ± 12.3 years old), particularly within the mental health domain. In the recent study by Fail et al. [23], only the intervention group that participated in the high-intensity aqua fitness class (as opposed to the moderate-intensity one) reported changes that were restricted to *physical quality of life* domain after 12 weeks with a training frequency of 2 times per week in a sample of older female individuals. These findings highlight the complexity of drawing direct comparisons between studies due to variations in methodologies and conceptual frameworks.

Positive changes were observed across all physical fitness parameters following the 10-week intervention, with notable improvements in lower-limb flexibility (moderate effect: ES = 0.85) and dynamic balance (moderate effect: ES = 0.73 and 0.65 for the right and left limb, respectively). This result may be attributed to the unique physical properties of water, specifically (i) the reduction in joint loading, which enables participants to perform movements with an extended range of motion, especially during lower-limb exercises [7], and (ii) the multidirectional body displacements and single-leg exercises characteristics of aqua fitness programs, which promote muscle engagement and strengthen stabilizing muscles essential for dynamic balance [44]. In addition to these moderate improvements, small gains were observed in upper-body muscle strength (ES = 0.20) and lower-body muscle endurance (ES = 0.29). While a systematic review and meta-analysis by Prado et al. (2016) [45] reported robust evidence of significant enhancements in muscle strength and endurance following aquatic exercise programs, the results of the present study align with the findings of Neiva et al. (2018) [46], who evaluated a water aerobics program of comparable duration. These findings suggest that healthy women aiming to achieve substantial increases in muscle strength may benefit from resistance training in aquatic environments or the use of in-water resistance tools (e.g., aquatic dumbbells and cuffs), which were not utilized in this study. Similarly, only small improvements were observed in mobility (ES = 0.48), likely due to the moderate intensity of the intervention and the program’s overall emphasis on general physical fitness parameters. All physical fitness parameters evaluated in this study were maintained regarding the effects of the cessation period, indicating that the positive adaptations achieved during the intervention were sustained over time. An exception was observed in lower-limb flexibility, which showed further increases during the cessation period. This outcome may be explained by the higher levels of physical activity commonly observed during summer months [47], especially among middle-aged individuals (*n* = 18 in this age group) [48], likely due to the *increased* engagement in outdoor recreational activities [49] (e.g., sea swimming and walking) that continue to challenge physical attributes such as range of motion. Given that most studies on the effects of a short-term training cessation period have focused on older individuals [50] or well-trained athletes [51], the findings of this study provide valuable insights. Specifically, they suggest that healthy women can engage in regular physical activity with confidence that unavoidable interruptions (e.g., illness or life events) are unlikely to result in significant declines in physical fitness. These results may serve as motivation for maintaining an active lifestyle.

The present study employed a real-world intervention approach, incorporating low-cost and straightforward physical fitness tests alongside measures of psychosocial functioning in a heterogeneous sample of healthy women, varying in chronological age and physical fitness levels. Aqua fitness programs are known to attract participants from diverse age groups and fitness backgrounds due to their low-impact nature, which makes them accessible and beneficial to a broad range of participants [18]. Additionally, the study included a comprehensive assessment of the participant’s physical fitness profile, which was repeated following 4 weeks of training cessation. The findings from this study provide valuable insights for fitness professionals and healthcare providers seeking to design more effective and sustainable exercise programs. Furthermore, the results validate aqua fitness as a viable training modality for promoting long-term health and fitness benefits.

Despite these contributions, several limitations should be considered. First, including a control group would have strengthened the ability to attribute the observed changes exclusively to the intervention and serve as a reference point for comparison. Nevertheless, the primary aim of this study was not to evaluate the effectiveness of the specific intervention. Second, although exercise intensity was monitored weekly using the RPE method, the movement velocity, a key determinant for any beneficial effects, was not controlled. Third, psychosocial functioning was only evaluated using questionnaires, a somewhat “convenient” tool with inherent limitations, such as response bias. Moreover, group dynamics, which could influence psychosocial outcomes through a potential sense of community, were not evaluated. In any case, integrating findings from a mixed-methods approach (i.e., combining questionnaires and interviews) could provide a more robust and nuanced understanding of psychosocial changes. Fourth, cardiorespiratory fitness, a critical parameter that determines the effectiveness of a physical activity program, such as in the case of aqua fitness, was not assessed. By not including this testing procedure, the authors aimed to reduce the waiting time by the participants, ensuring a more active engagement in the subsequent aqua fitness sessions while also improving overall satisfaction and adherence to the study. Finally, the potential changes in nutritional habits or physical activity levels during the intervention period, which could have influenced the results, were not controlled. Future research addressing these limitations would further enhance the validity and applicability of findings in similar contexts.

## 5. Conclusions

In conclusion, this study demonstrates that a 10-week aqua fitness program, performed three times per week for approximately 55 min per session at moderate intensity, is effective in improving the physical fitness profile of healthy women. However, these enhancements did not translate into significant improvements in psychosocial outcomes, likely due to the multidimensional nature of psychosocial constructs and the inherent challenges in their measurement. Finally, the physical fitness gains achieved during the intervention were maintained following 4 weeks of training cessation, highlighting the potential of this exercise modality to produce lasting benefits within this demographic.

## Figures and Tables

**Figure 1 healthcare-13-00334-f001:**
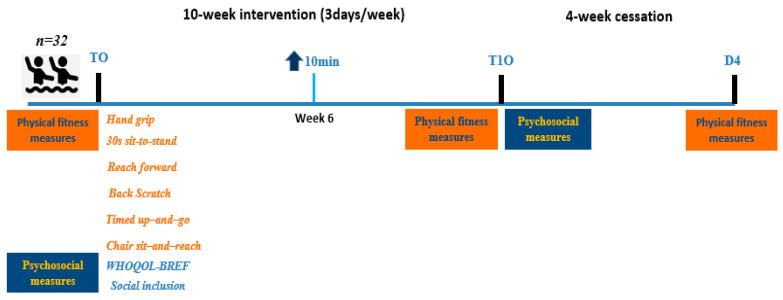
Schematic representations of the testing procedure. Note: T10 = end of the intervention; D4 = end of the training cessation period; Week 6 = 10 min increase in training duration.

**Table 1 healthcare-13-00334-t001:** A typical training session during the aqua fitness program.

Segments	Duration (Min)	Exercises	Brief Description
Warm-up	5		Forward and backward direction: leg action in a circular trajectory with arm movements in front, sideways, up, and down.
Main training program	35	Strengthening	Using wrist–ankle bracket, swimming pool round pair, aqua aerobic neoprene gloves, and swim noodles.
		Maintaining a vertical body position	Using alternating arm movements in various directions while holding the balance position.
		Arm and leg movements	Abduction, adduction, cross movements, and flexion in different positions (e.g., standing supine, prone, and side positions).
		Kicks	Performed in various directions rhythms and foot positions.
		Running	Alternating leg and arm movements with changes in direction.
Cool down	5		Stretching and relaxation at the edge of the pool.

**Table 2 healthcare-13-00334-t002:** Results of the physical fitness parameters and pairwise comparison between all testing points (mean ± SD).

	Training Week 0: (Τ0—Baseline)	Training Week 10: (Τ10)	Detraining Week 4: (D4)	Pairwise Comparison
Body mass	70.8 ± 12.6	70.12 ± 12.713.0	70.0 ± 12.6	-
Hand grip dominant arm (kg)	27.1 ± 5.0	28.8 ± 5.6	28.3 ± 6.9	T0 vs. T10, *p* < 0.001 T10 vs. D4, *p =* 1.000 T0 vs. D4, *p =* 0.294
30 sit-to-stand (s)	16.4 ± 4.2	17.7 ± 3.7	17.4 ± 3.1	T0 vs. T10, *p =* 0.019 T10 vs. D4, *p =* 1.000 T0 vs. D4, *p =* 0.117
Back scratch right (cm)	−3.4 ± 10.4	0.5 ± 9.5	0.3 ± 11.2	T0 vs. T10, *p =* 0.003 T10 vs. D4, *p =* 1.000 T0 vs. D4, *p =* 0.004
Back scratch left (cm)	−6.1 ± 11.4	−6.8 ± 10.1	−4.4 ± 11.8	T0 vs. T10, *p =* 1.000 T10 vs. D4, *p =* 0.031 T0 vs. D4, *p =* 0.289
Sit and reach right (cm)	1.6 ± 8.4	4.8 ± 8.0	7.8 ± 10.2	T0 vs. T10, *p =* 0.028 T10 vs. D4, *p =* 0.025 T0 vs. D4, *p <* 0.001
Sit and reach left (cm)	2.1 ± 6.8	5.1 ± 6.9	7.5 ± 7.9	T0 vs. T10, *p =* 0.001 T10 vs. D4, *p =* 0.022 T0 vs. D4, *p <* 0.001
Timed up–and–go (s)	6.5 ± 1.0	6.1 ± 1.1	5.7 ± 0.8	T0 vs. T10, *p =* 0.006 T10 vs. D4, *p =* 0.031 T0 vs. D4, *p <* 0.001
Reach forward (cm)	107.6 ± 8.7	113.3 ± 6.9	113.1 ± 7.0	T0 vs. T10, *p =* 0.001 T10 vs. D4, *p =* 0.197 T0 vs. D4, *p <* 0.001

## Data Availability

Data are contained within the article.

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
