# Peer review of "Impact of a 10-Week Aqua Fitness Intervention on Physical Fitness and Psychosocial Measures in Inactive Healthy Adult Women"

_healthcare, 2025, doi:10.3390/healthcare13030334_

Round 1
Reviewer 1 Report
Comments and Suggestions for Authors
1. Please add some background information in your abstract rather than only stating the purpose of the study.
2. line 97, "Age" should be replaced with "Aged"
3. It is unclear where the intervention took place and in which country. Please elaborate more on the characteristics of your population in the methods section.
4. I believe this is a quasi-experimental study rather than a cohort study as this program was administered in a pre-/post-test manner. Can you please clarify which study design was employed?
Author Response
Manuscript healthcare-3446895 - Revision 1
Healthcare
“Impact of a 10-Week Aqua Fitness Intervention on Physical Fitness and Psychosocial Measures in Healthy Adult Women”
Dear Editors and Reviewers,
Thank you very much for your helpful and constructive comments. Following the comments of the three reviewers, we have attempted to clarify and improve the manuscript. Enclosed is the revised version of the manuscript with all changes highlighted in red. This letter contains a point-by-point response to the reviewers’ comments.
We thank you for your consideration and hope that our responses will satisfy your requirements. We sincerely appreciate the time and effort taken in reviewing this manuscript and your valuable feedback which has helped to increase the quality of our manuscript.
Yours sincerely,
The authors

Reviewer 2 Report
Comments and Suggestions for Authors
Thank you for sending me this manuscript for review. The topic is interesting, but there are concerns that should be addressed first.
It would be better to write ""middle-aged " in the title instead of "adults"
It would be better to include an introduction to the abstract.
Why were only women involved in this study?
What was the reason for choosing 10 weeks of training and also 4 weeks of cessation?
In the participant’s section; when you talk about the GPower software; you should provide more information. For example; the effect size used and also the statistical method used to calculate the sample size.
What was the reason for choosing this age group? Wouldn't it be better to select older women? It would be better to mention this point in the limitations section.
Considering that you wrote in the inclusion criteria section; not participating in a regular physical activity program in the past 6 months; it would be better to mention the word "inactive women" in the title.
It is suggested that for each of the tools as well as the measurement process; a picture of the subjects be included.
Were the questionnaires on paper or on a mobile phone or laptop? Please provide more information on this.
Author Response

(The authors gave the same response as above.)

Reviewer 3 Report
Comments and Suggestions for Authors
I find the article very interesting, but I would like you to clarify a few issues before I proceed with a detailed review.
Introduction
The introduction is adequate, but citations are needed in some sections of the article, such as in the paragraph from lines 60 to 66. There are also missing citations in line 67; to which previous studies does it refer?
In the section on psychosocial benefits, it would be helpful to briefly define terms like "social functioning" and "psychosocial changes" for greater clarity and precision.
In lines 80-84, where the hypotheses are stated, I suggest highlighting them to make them more prominent for the reader.
Materials and Methods
I have a question regarding the study methodology. What would have happened if a different type of activity had been conducted instead of the intervention proposed? For example, strength training at a training center. Would similar results have been obtained? This raises questions about the validity of the intervention.
Furthermore, the reliability values of the questionnaires used to assess psychosocial dimensions are not specified; please include them.
It is unclear how the sessions were conducted. It would be useful to include a table with a brief description of the objectives and contents of each session.
Another issue is who carried out the intervention: was it the researcher? The psychosocial parameters might have been influenced by an external bias. Perhaps it would have been better if the same instructor who taught the participants six months prior had carried out the intervention. I would appreciate clarification on this matter.
Results
The results are appropriate based on the data analysis applied.
Discussion
The discussion is good, but I believe there are many surprises for the reader. It would be advisable to include many of the citations presented in this section in the introduction as well; otherwise, the reader might feel lost regarding the topic discussed.
Additionally, from lines 324 to 355, there are no citations, which may give the impression of poor-quality writing. Please restructure this paragraph, as it is too long.
Lastly, I suggest not including limitations in the discussion section, but rather presenting them in a separate section.
Author Response

(The authors gave the same response as above.)

Round 2
Reviewer 2 Report
Comments and Suggestions for Authors
Ok. Thank for the corrections!
Reviewer 3 Report
Comments and Suggestions for Authors
Dear Authors,
I want to congratulate you on the excellent work you have done in your revision. Your effort and dedication are evident in the quality of the manuscript, and I have no further suggestions for you.